# Predictors and triggers of incivility within healthcare teams: a systematic review of the literature

Sandra Keller [1], Steven Yule,[1,2,3,4] Vivian Zagarese,[5] Sarah Henrickson Parker[5,6,7]

¹Center for Surgery and Public Health (CSPH), Brigham and Women's Hospital, Boston, Massachusetts, USA
²STRATUS Center for Medical Simulation, Boston, Massachusetts, USA
³Department of surgery, Harvard Medical School, Boston, Massachusetts, USA
⁴Department of Clinical Surgery, University of Edinburgh, Edinburgh, United Kingdom
⁵Department of Psychology, Virginia Tech, Blacksburg, Virginia, USA
⁶Fralin Biomedical Research Institute (FBRI) at Virginia Tech Carilion, Roanoke, Virginia, USA
⁷Center for Simulation, Research and Patient Safety, Carilion Clinic, Roanoke, Virginia, USA

**Correspondence to**
Dr Sandra Keller;
sandra.keller@insel.ch

## ABSTRACT

**Objectives** To explore predictors and triggers of incivility in medical teams, defined as behaviours that violate norms of respect but whose intent to harm is ambiguous.

**Design** Systematic literature review of quantitative and qualitative empirical studies.

**Data sources** Database searches according to the Preferred Reporting Items for Systematic Reviews and Meta-Analyses guideline in Medline, CINHAL, PsychInfo, Web of Science and Embase up to January 2020.

**Eligibility criteria** Original empirical quantitative and qualitative studies focusing on predictors and triggers of incivilities in hospital healthcare teams, excluding psychiatric care.

**Data extraction and synthesis** Of the 1397 publications screened, 53 were included (44 quantitative and 9 qualitative studies); publication date ranged from 2002 to January 2020.

**Results** Based on the Medical Education Research Study Quality Instrument (MERSQI) scores, the quality of the quantitative studies were relatively low overall (mean MERSQI score of 9.93), but quality of studies increased with publication year (r=0.52; p<0.001). Initiators of incivility were consistently described as having a difficult personality, yet few studies investigated their other characteristics and motivations. Results were mostly inconsistent regarding individual characteristics of targets of incivilities (eg, age, gender, ethnicity), but less experienced healthcare professionals were more exposed to incivility. In most studies, participants reported experiencing incivilities mainly within their own professional discipline (eg, nurse to nurse) rather than across disciplines (eg, physician to nurse). Evidence of specific medical specialties particularly affected by incivility was poor, with surgery as one of the most cited uncivil specialties. Finally, situational and cultural predictors of higher incivility levels included high workload, communication or coordination issues, patient safety concerns, lack of support and poor leadership.

**Conclusions** Although a wide range of predictors and triggers of incivilities are reported in the literature, identifying characteristics of initiators and the targets of incivilities yielded inconsistent results. The use of diverse and high-quality methods is needed to explore the dynamic nature of situational and cultural triggers of incivility.

## Strengths and limitations of this study

► To our knowledge, this is the first systematic review on current empirical findings identifying predictors of incivility from both medical and nursing literature.

► To explore the predictors and triggers of incivilities, methods included quantitative and qualitative studies, which allowed an overview of the topic beyond methodological boundaries.

► Examining a wide range of predictors contributes to shed light on which predictors were already extensively investigated and for which predictors more empirical research is needed.

► Overall, the quality of the included studies was low and the conceptualisation of incivility and related terms based mainly on retrospective studies of study participants' perception; this is an inherent limitation to the review.

## INTRODUCTION

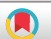Incivility among healthcare professionals has recently drawn increased attention in the medical world. The potential of incivility to jeopardise optimal patient care—and in turn patient safety, represents one of the major factors that led to their identification as a latent issue in healthcare.[1 2] Defined as behaviours that violate norms of respect but whose intent to harm is ambiguous,[3] incivilities are not typically in the scope of legal sanctions—despite their negative effects.[4]

Healthcare professionals themselves perceive an association between incivilities and decreased patient safety.[5] For example, a simulation study found a negative effect of rude behaviour on speaking up in medical students.[6] This result was supported by other simulation studies showing a decrease in communication after the expression of incivilities and also showing negative impact on performance.[7] In other domains, incivility showed negative effects both on well-being of employees and turnover.[8]

More than three-quarters of healthcare employees have witnessed incivilities by physicians and almost two-thirds incivilities

by nurses.[9] In another study, 85% of the nurses reported having personally experienced incivilities in the past year.[10] These findings outline the importance and prevalence of the phenomena and the need for additional efforts to reduce frequency and impact. The design of efficient interventions to reduce incivilities is closely tied to an accurate knowledge of the *predictors and triggers* of incivility in health teams. Predictors are not clearly articulated in the literature and have been explored in a piecemeal fashion. This literature review aims to provide a broad overview of the current empirical knowledge on predictors of incivility.

In this manuscript, we report the results of a systematic review on predictors of incivility in hospitals, including papers up to January 2020. Because a common characteristic of uncivil behaviours is the ambiguity around the intent to harm,[3 11] the review investigated closely related and often overlapping terms: incivility, rudeness, disruptive behaviours, interpersonal tensions and the disruptive behaviour part of unprofessional behaviours. These concepts describe impolite and rude conduct[12] and include overt behaviours such as yelling,[13] and racial or gender bias.[14] It also includes more subtle behaviours such as silences, rebukes,[15] gossip and displaced frustration.[16] Treating others like they are invisible or carelessness by colleagues can also be perceived as incivility.[17]

The medical, and in particular, the nursing literature also uses other terms such as verbal abuse (eg, accusing, blaming, yelling, insulting, humiliating, swearing),[13] horizontal or lateral violence (ie, violence across members of a same professional group) and bullying, a long-term form of lateral violence[18] to describe episodes of incivility or violence among health professionals. Because the mechanisms underlying more severe or long-term intrapersonal conflictual behaviours may differ from the ones underlying incivility, we restricted the focus of the present literature review on incivilities and low-intensity aggressive behaviours.

We examined empirical studies that report predictors of incivilities among healthcare teams in hospitals, including physicians, nursing and other professionals involved in patient care in hospitals. We investigated characteristics of both initiators and targets, their professional background and the situational and cultural predictors of incivilities.

## METHODS
The search for literature and the reporting of the results were conducted following the Preferred Reporting Items for Systematic Reviews and Meta-Analyses guidelines.[19] Quantitative and qualitative studies were included.

### Eligibility criteria
We included original publications of empirical studies focusing on predictors and triggers of incivilities among healthcare hospital teams. Studies conducted with medical or nursing students were included if they

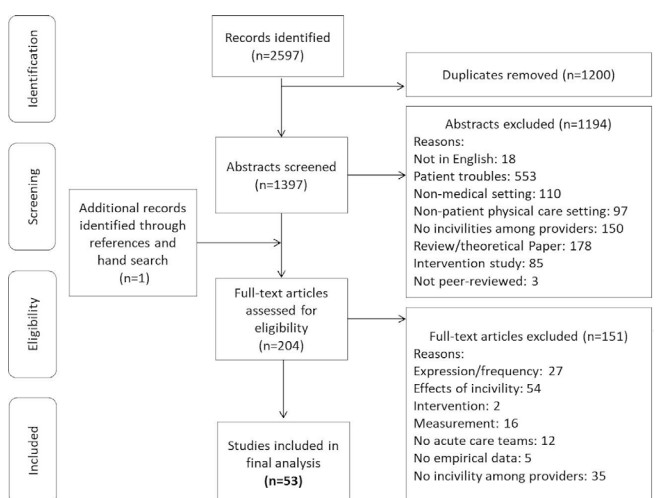

**Figure 1** Flow diagram of the selection process of studies included.

focused on clinical experiences of the students. Studies conducted in classroom educational settings were considered as not relevant because we aimed at capturing the dynamics of incivility in the clinical and patient care settings, where time pressure and stress are potentially higher. We included studies related to healthcare professionals working mainly in hospitals, with the exception of psychiatric hospitals. This decision was motivated by the potentially higher prevalence of patient incivility in psychiatric care settings, whereas the focus of this reviews is on incivility within healthcare teams. We set no restrictions in terms of year of publication and searched the full databases up to January 2020, but considered only papers published in English and in peer-reviewed journals with empirical findings related to predictors for incivilities.

### Information sources and search strategy
One author (SK) searched publications in four different databases: Medline, CINHAL, PsychInfo, Web of Science and Embase in January 2020. The search included incivility-related concepts combined with healthcare professions or major services in the hospitals where non-psychiatric patient care takes place. We followed a systematic search and inclusion-exclusion criteria (figure 1). The Medline database search strategy is included in online supplementary table 1. We hand searched the references for additional articles.

### Study records: data management and selection process
Publication records were independently extracted from the databases and transferred into an Endnote File. Duplicate articles were excluded. Publication records were then transferred from Endnote to a spreadsheet before coding. A multiple-choice menu was created to code the reasons of exclusion. In a first step, two reviewers (SK and SHP) independently assessed titles and abstracts of the articles for inclusion. All articles potentially reporting empirical original studies on predictors of uncivil behaviours were selected for full-text screening. Divergence in coding were

resolved by discussion. In a second step, two raters (SK and VZ) screened the full texts to identify studies meeting the inclusion criteria. Again, differences between the two raters were resolved by discussion within the rating team (SK, SHP, VZ). See figure 1 for a schema of the data management process.

### Risk of bias

The quality of quantitative studies was assessed with the Medical Education Research Study Quality Instrument (MERSQI) scale by one author (SK). The MERSQI scale is a validated tool originally designed to assess the quality of medical education publications; it is based on systematic ratings of the study design, sampling, type of data included, validity of measure instruments, data analysis and type of outcome reported.[20]

### Synthesis

The main goal of the review was to identify the predictors of incivility reported in empirical studies. We categorised the predictors of incivilities reported in the studies into five categories: (i) individual characteristics of initiators of incivilities, (ii) individual characteristics of targets of incivility, (iii) professional groups involved in incivility episodes, in terms of professional background and medical specialisation or hospital department, (iv) situational aspects and (v) cultural determinants. Specific concepts, methods and measurement tools used in the studies were also extracted (table 1).

### Patient and public involvement

It was not appropriate or possible to involve patients or the public in the design, conduct, reporting or dissemination plans of our research.

## RESULTS

The total number of studies selected was 53. We first present descriptive results about the studies, and then discuss their content. Content results are split into initiators, targets, medical specialties, situations and cultural and organisational characteristics.

### Descriptive results of the studies

#### Time frame

Studies meeting the inclusion criteria were published between 2002 and 2020. There was a sharp increase in the number of published studies in 2013, after that the number of published studies remained relatively stable, but on a low frequency level, with four to five published studies per year; since 2018, the number of studies again increased.

### Methodology of the included studies

Forty-four of the 53 studies included quantitative analysis and 9 were based on a qualitative design (table 1). Among the quantitative studies, the majority, 39 studies, relied on cross-sectional research design and used questionnaires. Other methodologies included analysis of prospective self-reports by the participants (events sampling),[21] data extracted from or collected in partly with an institutional electronic reporting systems,[22–24] data collected as part of a physician fitness to practice evaluation programme[25] or direct observations.[26]

Qualitative studies included four interview studies,[27–30] one observational study,[15] one study based on a combination of observations and interviews[31] and one qualitative analysis of reporting systems.[32]

### Quality of studies included

MERSQI scores, used to assess the quality of the quantitative studies, were relatively low overall, with a mean MERSQI score of 9.93, ranging between 6.5 and 14 on a scale from 5 (lowest possible MERSQI score) to 18 (highest possible MERSQI score) (details of the MERSQI scores for each study are available in online supplementary table 2). More recent publications showed higher MERSQI scores; we found a correlation of 0.52 ($p < 0.001$) between year of publication and MERSQI scores, see figure 2).

Methodological limitations were often similar across studies. First, many studies relied solely on participants' perceptions, with the exception of four studies based on the evaluation of a fitness to practice evaluation committee,[25] an expert committee examining the perspectives of multiple professionals involved in a same incivility event,[24] systematic observations[26] and an ethnographic observational study.[15] Second, most questionnaire studies reported low response rates, with a response rate below 50% in 28 studies. Third, nine studies described prevalence of disruptive behaviours and their triggers, but did not report more complex statistical analyses.

### Predictors of incivility

The results for each subcategory of predictors of incivilities are summarised and the situational and cultural predictors are presented in table 2.

#### Initiators of incivility

When asked about the main triggers of incivilities, healthcare professionals consistently mentioned personality as a major contributor to incivilities or that incivilities were initiated repeatedly by the same individuals.[27 29 30 33–36] One study showed that personality disorders were indeed more frequently diagnosed in physicians evaluated for disruptive behaviour than physicians evaluated for other issues (eg, sexual harassment).[25] No other study investigated specific personality characteristics of initiators of incivilities.

Evidence of demographic characteristics of initiators of incivilities was scarce, with one study exploring characteristics of uncivil physicians and two studies exploring the characteristics of uncivil nurses. The only overlapping result across the three studies was that initiators were more likely to be middle-aged or older than their targets.[22 25 28] Two studies found that initiators of incivilities were more likely to belong to the dominant racial

**Table 1** Studies included (n=53): settings, methods and predictors investigated

| Study | year | Country | Setting | Concept studied | Methods | Participants (N) | Focus | MERSQI score |
|---|---|---|---|---|---|---|---|---|
| **Physician to physician** | | | | | | | | |
| Pattani et al[30] | 2018 | Canada | Mixed: hospitals affiliated with a faculty of medicine | Incivility | Interviews | Faculty members (n=49) | Initiators Situation Culture | n/a* |
| Shetty et al[21] | 2016 | Australia | One ED | Incivility | Prospective self-reports of tone of phone conversations (tool designed by the authors) | Junior and senior physicians rotating or training in the ED (n=21 physicians, 714 phone consultations) | Target Profession Situation | 12 |
| Bradley et al[49] | 2015 | England | Mixed: three academic hospitals | Rude, dismissive and aggressive communication | Focus groups and questionnaires (probably designed by the authors) | junior doctors, registrars and consultants (n=606) | Profession Situation Culture | 7 |
| **Physicians to all** | | | | | | | | |
| Elhoseny and Adel[60] | 2016 | Egypt | Medical, surgical, ICU, anaesthesia, ED and pathology departments of one hospital | Disruptive behaviour | Questionnaire (based on the ACPE and QuantiaMD Survey[78]) | Physicians (n=120) | Situation Culture | 6.5 |
| Bansal[35] | 2014 | n/a | One tertiary care hospital | Disruptive behaviours | Questionnaire, developed by the authors | Doctors, nurses and technicians (n=614) | Initiators | 8 |
| Cochran and Elder[27] | 2014 | n/a – probably USA | OR | Disruptive behaviour | Interviews | Medical students, anaesthesiologists, residents, nurses and scrub techs (n=19) | n/a (open interviews) | n/a |
| Brewer et al[42] | 2013 | USA | Mixed: hospitals (68% of participants), and institutions | Verbal abuse | VAS Questionnaire (by Pejic, 2005[79], shortened 6-item version | New nurses (up to 6 years as a nurse) (n=1328) | Target Situation Culture | 9.5 |
| Finlayson et al[25] | 2013 | n/a – probably USA | Mixed: hospitals | Disruptive behaviour | Retrospective chart analysis of fitness-for-duty evaluation (Vanderbilt Comprehensive Assessment Programme) | Physicians (n=381) | Initiators Profession | 13 |
| Goettler et al[23] | 2011 | USA | Mixed: one academic hospital | Disruptive behaviour | Retrospective chart analysis of behaviours reported to the hospital system | Physicians (n=114) for 191 reported events | Initiators Profession | 10 |
| **All to physicians** | | | | | | | | |
| Klingberg et al[57] | 2018 | Switzerland | ED of one hospital | Incivility, bad manners | Questionnaire, developed by the authors | Physicians (n=50) | Professions | 9.5 |

Continued

**Table 1** Continued

| Study | year | Country | Setting | Concept studied | Methods | Participants (N) | Focus | MERSQI score |
|---|---|---|---|---|---|---|---|---|
| Birks et al[46] | 2017 | Australia and UK | Probably mixed: nurses recruited via heads of nursing schools | Workplace bullying | Questionnaire, SEBDCP survey (Budden et al, 2017[47]) | Australian (n=883) and UK (n=561) nurses students | Target Profession Culture | 10 |
| Budden et al[47] | 2017 | Australia | Probably mixed | Bullying and harassment | Questionnaire, SEBDCP survey, developed based on the work of Hewett (2010)[80] | Nurses students (n=888) | Target Profession | 10 |
| Small et al[10] | 2015 | USA | Probably mixed: different hospitals | Disruptive behaviours and verbal abuse | Questionnaire, developed by the authors | Nurses (n=2821) | Targets Professions | 9 |
| Elmblad et al[53] | 2014 | USA | OR and perioperative | Workplace incivility | Questionnaire, NIS (by Guidroz et al, 2010[81]) | Certified registered nurse anaesthetist (n=385) | Professions | 11 |
| Mullan et al[14] | 2013 | USA | Mixed: one hospital group | Disruptive behaviour | Questionnaire, developed by the authors | Medical interns (394) and attending physicians (40) | Target Profession | 10 |
| Lewis and Malecha[56] | 2011 | USA | OR, medical surgical, ICU, ED and women's services | Workplace incivility | Questionnaire: NIS (by Guidroz et al, 2007[82]) | Nurses (n=659) | Professions Culture | 10 |
| **Nurses to nurses** | | | | | | | | |
| Alkaabi and Wong[63] | 2019 | Canada | Mixed, probably many different hospitals | Incivility | Straightforward Incivility Scale by Leiter and Day (2013)[83], only the manager part | New graduate nurses (n=1020) | Culture | 11 |
| Arslan Yürümezoğlu and Kocaman[66] | 2019 | Turkey | Mixed: in two state academic/teaching hospitals | Incivility | Workplace Incivility Scale developed by Cortina et al (2001)[84] | Nurses (n=574) | Culture | 11 |
| Chang et al[45] | 2019 | South Korea | Mixed: three tertiary hospitals | Verbal abuse | VAS Questionnaire (Pejic, 2005[79]) | Nurses (n=378) | Targets Profession Culture | 12 |
| Tikva et al[67] | 2019 | Israel | Probably mixed, many different hospitals | Disruptive behaviour | Questionnaire developed by the authors | Nurses (n=567) | Culture | 10 |
| Keller et al[13] | 2018 | USA | Mixed: hospitals were the workplace of 75% of participants | Verbal abuse | Questionnaire: developed by Budin et al[43] | Early career nurses (n=1208) | Target Situation Culture | 12 |
| Smith et al[61] | 2018 | USA | Mixed: medical surgical or critical progressive care units in five hospitals | Incivility | Questionnaire: Workplace Incivility Scale (Cortina et al, 2001[84]) | Nurses (RN) (n=233) | Culture | 11 |
| Viotti et al[59] | 2018 | USA and Italy | Mixed: one hospital system in the USA and one hospital in Italy | Incivility | Questionnaire: co-worker incivility with scale adapted by Sliter et al (2012)[85] | US nurses (n=341) and Italian nurses (n=313) | Situation Culture | 11 |

Continued

**Table 1** Continued

| Study | year | Country | Setting | Concept studied | Methods | Participants (N) | Focus | MERSQI score |
|---|---|---|---|---|---|---|---|---|
| Kaiser[12] | 2017 | n/a | Mixed: acute and continuing care (unclear how many facilities included) | Incivility | Questionnaire: NIS (Guidroz et al, 2010[81]) | Staff nurses (n=237) | Targets / Profession / Culture | 10 |
| Boateng and Adams[28] | 2016 | Canada | Probably mixed: nurses recruited in two cities | Intraprofessional conflict | Interviews (one-on-one) | Nurses (n=66) | Initiators / Targets / Situation | n/a |
| Budin et al[43] | 2013 | USA | n/a | Verbal abuse | VAS Questionnaire (Pejic, 2005[79]) | Nurses (n=1407) | Target / Profession / Situation / Culture | 10.5 |
| Sellers et al[37] | 2012 | USA | Mixed: 19 facilities | Horizontal violence | Questionnaire: Briles' Sabotage Savvy Quiz[86] | Nurses (n=2659) | Target / Culture | 10 |
| **All incivilities and nurses' point of view** | | | | | | | | |
| Alshehry et al[38] | 2019 | Saudi Arabia | Mixed, wo government hospitals | Incivility | NIS developed by Guidroz et al (2010)[81] | Nurses (n=378) | Targets / Professions | 11 |
| Layne et al[58] | 2019 | USA | One hospital, level 1 trauma centre | Incivility | NIS (Guidroz et al, 2010[81]) | Nurses (n=414) | Professions | 9 |
| Minton and Birks[62] | 2019 | New Zealand | Mixed, different hospitals | Bullying/Harrassment | Questionnaire, SEBDCP survey, by Budden et al[47] | Nursing students enrolled in a bachelor programme (n=296) | Culture | 10 |
| Minton et al[48] | 2018 | New Zealand | Probably mixed, hospitals and other settings | Bullying/Harassment | Questionnaire, SEBDCP survey, by Budden et al[47] | Nursing students enrolled in a bachelor programme (n=296) | Targets / Profession | 9.5 |
| Ruvalcaba et al[40] | 2018 | USA | Probably mixed, in diverse hospitals | Incivility | Questionnaire, UBCNE tool (Anthony et al, 2014[87]) | Nursing students (n=975) | Targets | 10 |
| Nemeth et al[88] | 2017 | USA | Probably mixed, one academic hospital | Lateral violence | Questionnaire, the LVNS developed by the authors | Nurses, staff, managers (n=663) | Initiators / Situations | 9 |
| Addison and Luparell[52] | 2014 | USA | Probably mixed, in two rural hospitals | Disruptive behaviours | Questionnaire, developed by Rosenstein and O'Daniel[51] | 57 nurses (n=57) | Professions | 7.5 |
| Sliter et al[54] | 2014 | USA | n/a | Interpersonal conflict | Questionnaire, ICAWS (Spector and Jex, 1998[89]) | Nurses (n=172) | Profession / Culture | 11 |
| Veltman[55] | 2007 | USA | Labour and delivery in 56 hospitals | Disruptive behaviours | Questionnaire, developed by Rosenstein and O'Daniel[51] | Nurse managers (n=56) | Professions | 7.5 |
| McLemore[29] | 2006 | n/a | n/a | Workplace aggression | Interviews | Nurses (n=4) | Initiators | n/a |

Continued

**Table 1** Continued

| Study | year | Country | Setting | Concept studied | Methods | Participants (N) | Focus | MERSQI score |
|---|---|---|---|---|---|---|---|---|
| Riley and Manias[31] | 2006 | n/a—probably USA | OR, three hospitals | Tension and interpersonal conflicts | Ethnographic observations, group and individual interviews | OR nurses (n=11) | Situations | n/a |
| **All incivilities and all's point of view** | | | | | | | | |
| Rehder et al[68] | 2020 | USA | Mixed, 16 hospitals in one healthcare system | Disruptive behaviours | Questionnaire, developed by the authors | Healthcare professionals (n=7923) | Profession Culture | 12 |
| Chrouser and Partin[36] | 2019 | USA | OR in one academic medical training centre | Disruptive behaviour | Field notes from residency interviews | Medical students (n=42) | Profession Initiators Situations | n/a |
| Heslin et al[24] | 2019 | USA | Mixed, in one large tertiary medical academic centre | Disruptive behaviour | Reports on disruptive behaviours, from the perspective of the reporter and the involved party | Event-based analysis (n=314 event reports) | Professions Situations | 14 |
| Keller et al[26] | 2019 | Switzerland | OR, two academic hospitals | Disruptive behaviours/ tense communication | Observations (SO-DIC-OR) (Seelandt et al, 2014[90]) and questionnaires developed by the authors | Event-based analysis (n=340 observed events) | Professions Situations | 13 |
| Villafranca et al[39] | 2019 | Canada, USA, UK, Australia, New Zealand, India, Brazil, other | OR in different hospitals | Disruptive behaviour | Questionnaire, developed by Villafranca et al[39] | Anaesthesiologists, nurses, surgeons, other (technicians, medical students) (n=7465) | Targets Culture Professions Culture | 11 |
| Bae et al[44] | 2016 | USA | Probably mixed, one urban academic medical centre | Disruptive behaviour | Questionnaire, Johns Hopkins Disruptive Clinician Behavior Survey (JH-DCBS)[91] | Nurses, midwifes, CRNAs, physician assistants, MDs (n=1559) | Targets Professions Situations Culture | 10 |
| Hamblin et al[22] | 2016 | USA | Probably mixed, in a large hospital system with seven hospitals | Workplace violence | Retrospective chart analysis based on quantitative material | Perpetrators (n=185) for 199 violence incidents | Initiators Targets Professions | 11 |
| Berman-Kishony and Shvarts[33] | 2015 | Israel | Probably mixed, one medical centre | Disruptive behaviour | Questionnaire, developed by the authors based on focus groups and meetings | Nurses (n=76) and physicians (n=58) | Initiators Situations | 9 |
| Hamblin et al[32] | 2015 | USA | Probably mixed, in a large metropolitan hospital system with seven hospitals | Workplace violence | Retrospective chart analysis based on qualitative material | Violence and incivility incidents for which a catalyst could be identified (n=135) | Professions Situations | n/a |
| Walrath et al[50] | 2013 | USA | Mixed, in one hospital | Disruptive behaviour | Questionnaire, developed by the authors | RN, MDs, affiliates (n=1559) | Professions | 9 |

Continued

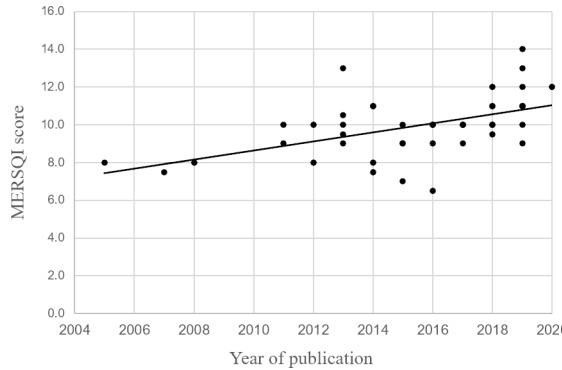

**Figure 2** Scatter plot and trend line of year of publication and Medical Education Research Study Quality Instrument (MERSQI) scores of the quantitative studies meeting the inclusion criteria of the current review.

group.[25 28] Physicians initiating incivility were predominantly males.[23 25 35]

### Targets of incivility

Fifteen studies included information on characteristics of healthcare professionals most likely to be targeted by incivilities. In figure 3, we present an overview of the empirical evidence.

*Gender* was the most investigated personal characteristic of targets of incivilities. Six studies conducted with healthcare professionals with different professional backgrounds found that females were more likely than males to be targeted.[21 27 37–40] Eight studies, also including different professional backgrounds, found no differences between females and males.[13 22 28 41–45] One study including nursing students in the UK and Australia, found that females were more likely to report incivilities in the Australian sample whereas in the UK, there was a trend that males were more likely to report incivilities.[46]

Research on which *age* groups were more likely to be targeted by incivility showed mixed results. Five studies found that younger health professionals were more likely to experience incivilities,[10 39 42 47 48] whereas four studies did not find differences across age groups.[13 14 22 43] Among nursing students, one study showed that older nursing students reported more incivility,[40] and another study found that nurses aged 25–27 years, but not aged 22–24 years, experienced more incivility than older nurses.[45]

Regarding *professional experience* (which is likely correlated with age), six studies showed that less experienced professionals were more likely to be targeted by incivilities.[14 38 39 44 45 49] Among nursing students, there was some evidence that advanced nursing students were more exposed to incivility.[40 46] One study showed no experience effect.[41] Overall, studies showed that less experienced team members were more often targets of incivility, but that different dynamics may operate during nursing education.

*Ethnical background* of targets was another characteristic often hypothesised to predict incivilities. Five studies found indeed that healthcare professionals with a non-dominant

**Table 1** Continued

| Study | year | Country | Setting | Concept studied | Methods | Participants (N) | Focus | MERSQI score |
|---|---|---|---|---|---|---|---|---|
| Rosenstein and Naylor[34] | 2012 | USA | ED, 20 different EDs | Disruptive behaviour | Questionnaire, developed by the authors | Physician, nurses, secretaries or clerks, ED technicians (n=237) | Personality Professions Culture Situations | 8 |
| Rosenstein and O'Daniel[9] | 2008 | USA | Mixed, in 102 hospitals | Disruptive behaviour | Questionnaire, developed by the authors | Physicians, nurses, administrative employees and others (n=4530) | Professions | 7 |
| Rosenstein and O'Daniel[51] | 2005 | USA | Mixed, in 50 hospitals | Disruptive behaviour | Questionnaire, developed by the authors | RN, physicians, administrators (n=1509) | Professions | 8 |
| Lingard et al[15] | 2002 | n/a | OR in one teaching hospital | Tension | Ethnographic observations | All OR team members (n=n/a) | Situations | n/a |

*MERSQI scores are only available for quantitative studies.
ACPE, American College of Physician Executives; ED, emergency department; ICAWS, Interpersonal Conflict at Work Scale; ICU, intensive care unit; JH-DCBS, Johns Hopkins Disruptive Clinician Behavior Survey; LVNS, Lateral Violence in Nursing; MERSQI, Medical Education Research Study Quality Instrument; n/a, not available; NIS, Nurse Incivility Scale; OR, operating room; SEBDCP, Student Experience of Bullying During Clinical Placement; UCBNE, Uncivil Clinical Behaviour in Nursing Education.

**Table 2** Situational triggers of incivilities in healthcare teams

| Study | Situation | |
|---|---|---|
| Brewer et al[42] | More physician abuse associated with fewer nurses working than scheduled. | Workload |
| Boateng and Adams[28] | If heavy work responsibilities, minority nurses reported conflicts about who did what (expertise). | Work responsibilities |
| Hamblin et al[32] | Work behaviour: unprofessional behaviour, duties and responsibilities, methods of care, poor performance.<br>Work organisation: conflicts about tasks and procedures, organisational constraints, interdependence between the workers. | Communication/teamwork<br>Patient safety<br>Work responsibilities<br>Organisational constraints |
| Nemeth et al[88] | Most highly causal explanation was stress related to inadequate staffing or resources, followed by societal decline in civil behaviour. | Workload |
| Keller et al[13] | Organisational constraints predicted more incivility; no effect of quantitative workload. | Workload (no effect)<br>Organisational constraints |
| Pattani et al[30] | Infrequent interactions. | Lack of familiarity |
| Viotti et al[59] | Workload as a predictor of incivility only in the USA but not in the Italian sample. | Workload (in one of the study samples) |
| Berman-Kishony and Shvarts[33] | High workload is the second most frequent cause reported, followed by poor communication, distrust and disrespect. | Workload<br>Communication/teamwork |
| Budin et al[43] | Higher levels of verbal abuse perceived by nurses as associated with: fewer nurses working than scheduled (staffing shortfalls), less perceived distributive and procedural justice, less promotional opportunities, more organisational constraints, higher quantitative workload. | Workload |
| Cochran and Elder[27] | In the operating room, incivility was associated with: unfamiliar teams or trainees, something goes wrong during the operation, when there are differences in opinions with the surgeon while planning the operation. | Familiarity<br>Workload or patient safety |
| Rosenstein and Naylor[34] | Delays, inadequate staffing and poor communication were rated less frequently than personality and attitudes. | Workload<br>Communication/teamwork |
| Riley and Manias[31] | Time: questioning judgement time, controlling speed, estimating surgeon's time, different perceptions of time. | Time |
| Elhoseny and Adel[60] | Workload as first root cause (reported by 35%), 15% reported compensation-related factors. Other: non work-related situations (12%). | Workload<br>Non-work-related factors |
| Bradley et al[49] | Doctors describing the situations in which they are rude: high workload, patient safety compromised, hierarchy. | Workload<br>Patient safety |
| Lingard et al[15] | Time, resources, roles, safety and sterility, situation control. | Communication/teamwork<br>Patient safety<br>Time |

**Table 2** Continued

| Study | Situation | |
|---|---|---|
| Bae et al[44] | Triggers of disruptive behaviours at the interindividual level (eg, questioning providers about care, lack of teamwork, staff diversity) and intrapersonal level (eg, lack of competency, fatigue) related to experienced disruptive behaviours. Among nurses only (not physicians) organisational triggers (pressure from high volume, overload, unresolved issues unit culture) were also predictors of disruptive behaviours. | Workload Communication/teamwork Patient safety Fatigue |
| Shetty et al[21] | Consultations with requests for investigations. | Request |
| Heslin et al[24] | Patient factors mentioned as triggers (eg, challenging anatomy), technical and environmental factors, organisational factors, stressors (individual or team). | Workload Communication/teamwork |
| Chrouser and Partin[36] | Patient factors mentioned as triggers (eg, challenging anatomy), technical and environmental factors, organisational factors, stressors (individual or team). | Communication/teamwork Organisational constraints Task difficulty/stress |
| Keller et al[26] | Collaboration and task-related issues were clearly more frequent sources of tensions than relationship issues or disagreement about the task. | Communication/teamwork Task difficulty/stress |
| Rehder et al[68] | Disruptive behaviours correlated with poorer experienced teamwork, lower job satisfaction and lower perception of management. | Communication/teamwork |

ethnical background or non-native speakers in the country where the study was conducted were more likely to experience incivilities,[27 28 44 46 48] whereas four studies did not find differences across ethnic groups.[13 39 43 47] Of note, two studies found contrasting results with non-native speakers reporting less incivility,[40 48] yet in one these studies, non-native speakers were also unsure about identifying the concept of incivility.[48]

Few studies focused on nurses' educational background[10 13 38 41 44] (eg, diploma vs baccalaureate),[38] shift

**Gender**
hypothesis: females > males [1]

**Age**
hypothesis: younger > older [2]

**Experience**
hypothesis: little experienced > experienced [3]

**Ethnicity / language**
hypothesis: non majority groups > majority [4]

Note. The size of the bubble represents the number of studies included that support the hypothesis, showed no differences between group respectively showed differences in the opposite direction

[1] Female healthcare professionals may experience more incivility

[2] Younger healthcare professionals may experience more incivility

[3] Healthcare professionals who have less work experienced may experience more incivility

[4] Healhcare professionals who belong to a visible ethnic minority group or are no native speaker may experience more incivility

**Figure 3** Strength of current empirical evidence on the association between characteristics of healthcare professionals and exposure to incivility.

type[13 42] or job tenure.[22 44] Cross-sectional studies investigating the association between psychological states such as work satisfaction and incivility are scarce and do not allow to identify consistent results.[13 43]

### Professional background and medical subspecialties

Results of the studies included allowed exploration of potential differences in the prevalence of incivilities across medical professions and medical domains. We first report differences across professional backgrounds, for example, nurse and physicians and second, we report comparisons across medical domains (eg, operating room (OR) vs intensive care unit (ICU)).

### Professional backgrounds

The most often examined research question pertained to the prevalence of incivilities in physicians and nurses, and studies investigated the most likely instigator of incivilities among professional groups.

### Perception of physicians

In one study, physicians perceived other physicians as the most frequent initiators of incivilities[14] and in another study, physicians perceived incivility by other physicians as incivilities having the most negative impact.[50] Medical interns reported nurses rather than physicians as most frequent initiators of incivilities.[14] In one study, results were less clear, with physicians perceiving about half of the incivilities initiated by nurses and the other half initiated by physicians.[51] Nevertheless, slightly more studies

reported that physicians are the primary source of incivilities to other physicians after training completion.

## Perception of nurses

A majority of studies (seven) found that nurses perceived other nurses as the most frequent or most negative source of incivility,[10 50 52 53] three studies were conducted with nursing students.[46–48] Four studies reported contrasting results, with physicians perceived as the most frequent source of incivilities by nurses[38 51 54] or nursing managers.[55]

## Studies including professionals from a variety of backgrounds

Not surprisingly, studies that surveyed diverse medical professionals found mixed results. One study found that physicians were most frequently initiators of incivility,[9] whereas another study reported similar rates of incivilities by nurses and physicians.[34] Two studies based on institutional reports found that nurses were more often involved in incivility episodes compared with other professions.[22] Of note, one of these studies did not include most incivility episodes reported by physicians.[22] Three OR studies showed contrasting results, with attending surgeons more likely than the other OR healthcare professionals to initiate uncivil episodes.[24 26 36]

Five studies focused on the professional groups most likely to be targeted by incivilities. These studies found that nurses or scrub technicians,[26 39 44 51] and in general, professions associated with less power in the medical hierarchical system[27]—more junior surgeons in one study[26]—were more frequently targeted by incivilities.

## Medical specialties

We addressed the question regarding the prevalence of incivilities across specific medical specialties. *Surgery* or surgical subspecialties appeared in five studies as one of the domains with the most incivilities, compared, for example, with paediatric or emergency departments (EDs),[12] family or internal medicine doctors,[25] the ICU or medical-surgical units[56] and other specialties outside radiology and cardiology,[49] with professionals spending more time in the OR reporting higher incivility levels.[39] One survey with ICU physicians found contrasting results, showing that surgical specialists were less likely to be uncivil to ICU physicians as compared with non-surgical specialists.[57] In the same vein, a study found that interactions with surgeons were rated by ED physicians similarly as interactions with other specialists.[21] Interestingly, in these two latter studies, surgeons were likely to work in other settings than the OR when they interacted with their medical colleagues.

In two studies, *radiology* appeared to be the specialty associated with the most incivilities. In one study, radiology was followed by general surgery, neurosurgery, cardiology and other specialties[49] and in the other study radiology was compared with medical, surgical and other specialties.[21] One study found contrasting results, with radiology as one of the medical domains with the least incivility, for example, compared with surgery,

cardiology, trauma and other potentially higher risk specialties.[23] Other medical domains that were associated with more incivilities were *obstetrics*[12 23]—with one study showing contrasting results,[38] long term-care,[12] the ED, ICU, cardiology,[23 52] whereas a study found that nurses working in the ICU reported the least incivilities compared with other nurses.[43] However, two studies did not find different perceived incivility levels when comparing general, intermediate and ICU, specialty care and nursing clinical support,[58] respectively general ward, ICU, emergency room and OR.[45]

Three studies that included physicians found that incivilities were more likely during *collaboration with other departments* compared with participants' own department,[23 49 57] suggesting that intergroup dynamics may also impact incivility. In one of these studies, contradictory results were found for nurses who reported more uncivil behaviours initiated by physicians within their own department than initiated by physicians external to their own departments.[23]

## Situational influences on incivilities

There is evidence that medical professionals report specific situations as fertile grounds for incivilities. We identified seven different situational triggers investigated in different studies and present these results in table 2.

*High workload* was the most often mentioned trigger of incivilities, reported in ten studies. One questionnaire study did not find an effect of workload, and another study found an effect of workload only in a sample of US nurses but not in a sample of Italian nurses.[59] The second most frequent situational factors identified as trigger of incivilities are related to the non-technical skills of *coordination, communication and teamwork* (eg, poor communication, lack of teamwork), reported in nine different studies. *Patient safety concerns* or poor performance were other factors triggering incivilities reported in three different studies based on ethnographic observations,[15] retrospective chart analysis[32] and questionnaires and focus groups.[49] Two studies found that situations in which healthcare professionals who experienced *heavy responsibilities* may be more prone for incivilities. In two studies conducted in the OR, *time* management and negotiations were triggers of tense situations.[15 31]

Team composition was also investigated as a potential trigger of incivility, with *little familiarity* among team members perceived as enhancing incivilities.[27 30] Finally, *organisational constraints*, defined as factors preventing employees to perform their task efficiently (eg, because a lack of resources), were perceived as a potential catalyst of incivilities,[13 32 36] as were task difficulties and stress.[26 36]

Some other situational factors investigated by a single study and contributing to incivilities in healthcare teams were fatigue,[44] personality conflicts,[24] the reason for the interaction, that is, request for medical investigations,[21] compensation or non-work-related factors.[60]

## Culture and organisation's characteristics

The relationship of culture, organisation of the department, the hospital or of countries to uncivil behaviour were investigated by different studies. We included results of studies that did not directly measure culture but closely related concepts, such as the impact of department leaders and studies comparing samples of participants working in different countries.

*Leadership* was associated with incivilities in several studies. Four studies investigating nurses found that the nurses managers' skills to handle incivilities[43 49 56 61] or setting the right tone[62] was a protective factor against incivilities. A study with physician faculty members found similar results, with participants pointing to the lack of reaction of leaders in handling less severe incivilities.[30] Furthermore, transformational[12] or authentic[63] leadership were found to be protective of incivilities whereas lack of leadership was associated with increased perceived incivility[44]; none of the studies provided data on how transformational leaders contribute to reduced incivility levels. Only one cross-sectional study did not find an association between perceived supervisor support and incivility.[13]

Workplace culture also seems to influence incivilities. For example, three studies found that nurses working in a magnet hospital, a label recognising the quality of nursing care and the professional development of the nursing workforce,[64] were less likely to experience incivilities. Only one study failed to find an effect[13 65] and one study found an association between incivility and private founded hospitals.[39] In three further studies that were conducted with physicians,[27 60] respectively with a mixed sample of physicians and nurses,[34] the authors found evidence that culture and training contribute to incivilities, suggesting that uncivil behaviours are learnt and fostered during physicians' training. Furthermore, a positive work culture and support from colleagues or the organisation[13 43 61 66–68] and a diversity climate[54] were associated with decreased incivilities in seven studies, without evidence of divergent results. In one study, distributive justice, but not procedural justice, was also associated with decreased incivility levels.[13]

Few studies focused on the impact of the countries' cultures on incivilities. Two studies, conducted with nurses, included samples from different countries. One found that the prevalence of incivilities was higher in the USA compared with the Italian nurse sample. The other study compared Australian with UK nurse students and found that Australian nurse students reported more incivility.

## DISCUSSION

This systematic review reports the current state of research related to triggers of uncivil behaviour, reporting consistent and inconsistent findings. Although the interest for this topic has been present for several years in the medical field, the number of studies reporting empirical work only recently started to increase. In addition, the quality scores for most studies, as assessed by MERSQI criteria, were comparable to other samples,[20] with only three quantitative studies and one qualitative study relying on other measurement methods than perceptions of the study participants. An important result of this review is the need for more empirical research of high quality.

Nevertheless, the existing studies cover a wide range of factors that underlie expression of incivility at work. These predictors or triggers of tensions range from the intrinsic characteristics of the people involved in incivility episodes to situational or cultural aspects influencing the emergence of incivilities. Existing models of incivilities in healthcare teams already include many of the triggers identified empirically, for example, the model of triggers of incivilities in the OR presented by Villafranca *et al*[69] that describes intrapersonal, organisational and interpersonal factors. However, they are not studied in a systematic way.

Studies investigating *initiators of incivilities* support the influence of personality on uncivil behaviour, sometimes described as 'bad apples'.[27] However, most of these studies are based on perceptions of study participants. Relatively few studies focused on initiators' perceptions and explored their motivations and interactional context, beyond personality.

Overall, the review shows that demographics of *targets* are not consistently related to incivilities. Although explored by 15 studies, it was not possible to identify consistent gender differences and specific age and ethnic groups as particularly likely targets of incivilities. However, the studies available on the association between work experience and incivilities show that more experience, often associated with a higher hierarchical status in the organisation, is associated with decreased experience of incivilities. This indicates that higher task proficiency, and higher status, may be protective factors. This finding is in line with the experience of physicians who observed that they were treated with more respect after their promotion to consultant compared with earlier stages of their medical career.[49]

In terms of *professional background* of tension initiators, the dynamics appeared to be more complex than could be expected. Results showed more evidence of incivilities within similar professional groups, as compared with interprofessional incivilities. Whereas this result is not surprising for physicians, it shows that nurses, rather than physicians, were, in most studies, reported as more likely to initiate incivilities. Of note, most studies did not measure nor control for the frequency of interactions within, and between, professional groups; this is an important potential bias. In addition, most studies are based on the perception of a specific professional group which may also be a source of bias.[70] The studies also failed to identify consistent differences among medical specialties, with the exception of surgeons during their work in the OR. This result may be explained by the more stressful work conditions, the closer cooperation and the higher risk tasks performed.[23]

Different *situational* aspects influence incivilities in healthcare teams, with workload, communication and teamwork as most important factors, followed by patient safety issues as compared with other predictors. Among *cultural* factors, leadership and support among the group as well as working in a hospital recognised for excellence in nursing care were among factors recognised as protecting against high incivility levels. Thus, these results suggest that rather than universal professional cultures, local dynamics in specific work situations, departments and hospitals may influence incivilities and should be considered.

Overall, the methodological quality was relatively low for many of the studies reviewed. Methods such as prospective and systematic observation of uncivil interactions[15 21 26] or relying on hospital surveillance systems[22 24 32] are rare. Even situational triggers of tensions which need to be studied specifically were investigated with cross-sectional survey studies. However, given the only relatively recent interest in this topic, it is important to note that some of the studies included in the review belong to the very first studies that focused on incivilities in healthcare teams. Thus, methodological weaknesses may be offset by the pioneering character of the work, and more recently published papers showed better methodological quality.

## STUDY STRENGTHS AND LIMITATIONS
### Strengths
One strength of the study was that we included papers based on different methodological approaches to answer the question of the systematic review. This approach allowed to assess similar research questions of studies relying on different methodologies. In addition, this more inclusive approach allows a more extensive overview of the topic.

Because teamwork in healthcare teams is inherently multidisciplinary, we included research conducted with nurses or a mixed population that was often done in nursing science as well as research conducted with physicians, often initiated by physicians. Furthermore, the search process revealed the impressive number of theoretical or position papers (183) on incivilities much more than empirical studies. The high number of theoretical papers is an indicator for the interest in the topic. To understand the phenomenon and what leads to incivilities, there is an urgent need for more empirical research, and in particular research that goes beyond questionnaire studies. Only empirical research can inform the conceptualisation and the understanding of processes triggering incivilities within healthcare teams.

### Limitations
A limitation inherent in the topic of incivility is the conceptualisations of incivilities and related behaviours are subjective, because the intent to harm is per definition ambiguous.[3] It is thus important to underline that studies that investigate incivility based on perceptions (ie,

questionnaire studies) cannot claim to measure incivilities and their triggers beyond participants' perceptions. However, recent studies are promising, showing that perceived incivility can be efficiently assessed with validated tools (see Harris *et al* for a review)[71] and methods relying on systematic analysis of institutional reports[24] or observations[26] are emerging.

The few studies focusing on the analysis of specific uncivil events rather than perceptions of those events indicate that uncivil behaviour is a complex phenomenon, and much more complex that one initiator behaving in an uncivil way towards a target.[15 23] We did not include conflicts in our search strategy, although conflict behaviour can be uncivil. Conflicts are traditionally defined as caused by divergent opinion on the task or process or caused relationship issues and are of longer term.[72] Yet, conflicts situations may well underlie uncivil episodes, and further analyses of conflicts in healthcare teams may also contribute to the understanding of uncivil episodes in this context.[73 74] Similarly, studies that included terms such as horizontal violence, lateral violence, bullying or other forms of aggression without reference to one of our search terms were not included. This allowed to focus the review specifically on less severe forms of rudeness. Yet, there is currently a lack of consistency on the definition of terms related to rude behaviours in the literature.[18 75] We thus cannot exclude that our search strategy did not allow to capture studies that relied on terms usually describing intentional intent to harm (eg, aggression)[75] and whose definitions widely overlapped with incivility in individual works.

## CONCLUSION
Given the known impact of incivilities on both patient care processes[7] and healthcare professionals' health,[76 77] the need for efficient interventions to reduce incivilities in healthcare teams is likely to increase. Such interventions need to be based on empirical evidence. The present systematic review showed that most studies investigated general characteristics of initiators and targets of incivilities. Situational aspects that foster incivilities are clearly understudied, so we may underestimate the probability that incivilities are a result of coordination problems. Further studies should concentrate on these situational triggers (cooperation, task requirements). Future incivility research in the medical field also needs to adopt higher quality methods than current studies. Only if these two conditions are satisfied can empirical results then inform the design of interventions to reduce incivility and the potential harm to providers and patients. Interventions at the organisational level are particularly likely to benefit from this research since healthcare organisations can influence to a certain degree the design of work processes, leadership within departments and cultural aspects that tackle rather than promote incivility.

**Acknowledgements** The authors would like to thank Rita McCandless and Erin M. Smith, PhD, for their technical support and advice about the library resources and giving us access to the full texts that could not be retrieved by the authors. The authors would like to thank Franziska Tschan, PhD for her critical review of an earlier version of the manuscript.

**Contributors** Study design: SK, SHP and SY. Data analysis: SK, VZ and SHP. Drafting the work or critically revising it: SK, SY, VZ and SHP.

**Funding** This work was sponsored by a grant of the Swiss National Science Foundation (grant number P2NEP1 178574).

**Competing interests** None declared.

**Patient and public involvement** Patients and/or the public were not involved in the design, conduct, reporting or dissemination plans of this research.

**Patient consent for publication** Not required.

**Provenance and peer review** Not commissioned; externally peer reviewed.

**Data availability statement** All data relevant to the study are included in the article or uploaded as supplementary information.

**Author note** Sandra Keller currently works at Bern University Hospital (Inselspital), Switzerland.

**ORCID iD**
Sandra Keller http://orcid.org/0000-0003-3229-9003

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
