## [Reviewer comments · BMJ Open]

ARTICLE DETAILS

TITLE (PROVISIONAL)	Predictors and triggers of incivility within healthcare teams: A systematic review of the literature
AUTHORS	Keller, Sandra; Yule, Steven; Zagarese, Vivian; Parker, Sarah

VERSION 1 – REVIEW

REVIEWER	Stefano Bambi Careggi University Hospital, Florence (Italy)
REVIEW RETURNED	21-Nov-2019

GENERAL COMMENTS	Dear editor, thank you for the opportunity to review the paper entitled “Predictors and triggers of incivility within medical teams: A systematic review of the literature”. This issue is very important and actual for the internal working climate in the healthcare professional teams, and it deserves special attention from the readers. The paper is well written. However, my major concern is related to the search strategies used by the authors, since the lack of some (in my opinion) fundamental keywords. However, since the strong methodology used by the authors, I think that they could easily improve this paper to make it suitable for publication. My personal comments are attached below Title I find the title a little bit misleading since the readers could think that the paper is centered only on doctors...I suggest changing the title including the words “healthcare professionals” Abstract The abstract is well structured, even if it does not follow all the indications included in the PRISMA check-list. However, it is clear and exhaustive. Article summary Well written and clear. Introduction Line 38: please provide a definition for medical team: does it include only physicians or physicians and other healthcare professional (eg. nurses)? Lines 47-56: please provide an explanation related to the decision to not include in your research, condition almost overlapping to incivility, as lateral (horizontal) violence, and harassment, since lots of the cited behaviors cannot be excluded from the intent to harm, with a high level of certainty.
--

	Methods This section is well written and rigorously drafted. I've only some concerns about the keywords used by the authors for their literature searching. The definition and the conceptuality of "workplace incivility" in the working settings are often blurred as other "negative behaviors" that can be displayed in "abuse", "hostility", "harassment", "lateral violence", and "bullying" terms. Since many of these keywords are reported in the "concept studied" column in table 1, I think that at least these keywords should have been included in the databases searching, and later, the retrieved papers should have been assessed for the presence of inclusion/exclusion criteria. Moreover, the authors included the category of medical students, but not the students in other disciplines, as the student in nursing. Lastly, the authors should clarify why they did not explore the healthcare professionals working incivilities in outpatient settings, out of hospital healthcare settings, and educational (universities) settings.
--	--

REVIEWER	Amith Shetty Westmead Institute for Medical Research Sydney Australia
REVIEW RETURNED	09-Dec-2019

GENERAL COMMENTS	Thanks for undertaking a thorough systematic review on a topic very pertinent to care delivery worldwide atm! Comments have been attached to the comments on the file attached. Broadly -in abstract focus on search strategy and findings rather than discussion, highlight key findings across all papers and state it clearly the manuscript needs a major revision and addition of data table to reduce the results section. Currently the results section is extremely long and repetitive and should be culled significantly to focus on key findings e.g. gender, age, position and specialty trends. - The reviewer provided a marked copy with additional comments. Please contact the publisher for full details.
---

REVIEWER	Reena Pattani St Michael's Hospital - Unity Health Toronto and University of Toronto Department of Medicine Canada
REVIEW RETURNED	09-Dec-2019

GENERAL COMMENTS	Thank you for preparing this manuscript on an interesting and urgent topic. I really enjoyed reading this and think it has a lot of potential to contribute to the existing evidence base on this topic. A few points to consider: 1) Abstract:  - The risk of bias assessment should also be included in the abstract with reference to the MERSQI tool. - The use of "primary and secondary outcomes" is explicit in the abstract, but not in the methods. 2) Methods:  - It may be beneficial to specify why only English language studies were included (if for feasibility reasons for example, which is an acceptable reason). The dates of coverage for the databases should be explicitly stated. The exclusion of psychiatry patient care contexts should be in the exclusion criteria (though I wonder if it is
---

	of merit to include them, and if they are excluded - clear reasons why should be provided).  - Who conducted the search of the databases? Was it a librarian? If so, this should be specified. Keywords used for the search can be included. - For steps 1 and 2 in the data abstraction, was a pre-made data abstraction form created? If so, was it pilot tested? It could be referenced and included as an appendix. - Following the statement "Divergence in coding were resolved by discussion" - was this discussion between the two independent reviewers or the entire research team? - The total number of studies selected should not be reported in methods, only in results. - Was the risk of bias assessment done independently by two investigators? If so, how were differences reconciled? - There is no risk of bias for the qualitative studies even though these comprised almost 20% of the included publications. You can consider using the Grades of Recommendation, Assessment, Development, and Evaluation–Confidence in the Evidence from Qualitative Reviews or another tool. 3) Results:  - I do not think the first paragraph in this section is required (beginning with "we will first present"), especially since the mode of synthesizing data was just outlined in the methods. - The first section of the medical specialties paragraph was a bit hard to interpret owing to a double negative: "In the same vein, a study found that interactions with surgeons were not rated more negatively by ED physicians than were interactions with other specialists". 4) Tables:  - Table 1 is very informative. I wonder if within each category of grouped studies, it is worth listing studies by MERSQI score? Is it possible to include the year of publication with the author name in column 1? I only suggest this because your discussion hypothesized that earlier studies may be of lower methodological quality, so it would be telling to see if year of publication correlated with MERSQI score. General comments:  1) I see that "type of data" is a MERSQI domain and that assessment by study participant gets scored lower than objective measurement, but I wonder if this is a limitation of the MERSQI tool in its application to studies on incivility, which, as a social construct, is reliant on the perceptions of individuals? A recent scoping review of tools used to measure incivility identified the short negative acts questionnaire (S-NAQ) as the best suited tool to utilize in healthcare settings (A Scoping Review of Validated Tools to Measure Incivility in Healthcare Settings. Harris et al 2019) - but the items in this tool also seem like they are based on participant perception. 2) Is there a way to correlate the MERSQI scores with high / moderate / low risk of bias? Or Good / moderate / fair quality of studies? If so, it might make it reader friendly to report how many studies fell into each category. 3) There were some grammatical errors in the paper that need to be reviewed carefully.
--	---

REVIEWER	Diana Layne Medical University of South Carolina
REVIEW RETURNED	11-Dec-2019

GENERAL COMMENTS	Thank you for the opportunity to review your manuscript on this very important topic. Overall I enjoyed reading this work. There are a few minor typographical errors that should be corrected. Further, I would recommend including information as to the rationale for including students in your sample. Potential differences exist in the sources of incivilities experienced by student learners, as sources could be faculty versus others within the hospital. Also, did you consult a medical librarian to develop your search criteria? If so, I would revise to include that information. Finally, Figure 2 on page 7 seems redundant I recommend removing from the manuscript. - The reviewer provided a marked copy with additional comments. Please contact the publisher for full details.
---

REVIEWER	Jayna Holroyd-Leduc University of Calgary, Canada
REVIEW RETURNED	23-Dec-2019

GENERAL COMMENTS	This is a high quality systematic review that follows the PRISMA criteria and also assesses the quality of included quantitative studies. It addresses an important issue - incivility in medicine and an exploration of what may predict or trigger this behaviour amongst practitioners. Incivility has been shown to negatively impact quality of patient care. Unfortunately, the quality of the literature included in this systematic review is poor and many unanswered questions remain. That being said, the authors attempt to examine key factors that one might think would predict incivility - focusing on individual, cultural and situational factors. This systematic review could be strengthened by considering the following:  1) The search is over 1 year old and therefore should be updated 2) The databases searched did not include Embase. To ensure key articles are not missing, Embase should be searched. 3) It may be helpful to add a table that summarizes the quantitative data, in terms of the number of positive and/or negative studies for the various predictive factors looked at. 4) The presentation of the qualitative data is less clear - more discussion of these studies and their findings may strengthen the manuscript. 5) When looking at gender, the authors don't appear to have separated out studies that focus on physicians from those that focus on nurses. Given that nursing is typically female dominated, the authors may want to look at the findings related to gender in only those studies that involved physicians. 6) The discussion could be strengthened by starting to explore potential strategies that could be implemented around the factors that were found to impact incivility (e.g. leadership factors).
---

VERSION 1 – AUTHOR RESPONSE

Reviewer: 1

I find the title a little bit misleading since the readers could think that the paper is centered only on doctors...I suggest changing the title including the words "healthcare professionals"

Response: We adapted the title of the manuscript accordingly and changed throughout the paper to healthcare teams or healthcare professionals. Thank you for this remark.

The abstract is well structured, even if it does not follow all the indications included in the PRISMA check-list. However, it is clear and exhaustive.

Response: In line with the comment of the editor, we adapted the structure of abstract.

Line 38: please provide a definition for medical team: does it include only physicians or physicians and other healthcare professional (eg. nurses)?

Response: We thank the reviewer for the comment on this important aspect. We clarified our definition in the last paragraph of the introduction. In the revised version, we use the term healthcare professionals/teams rather than medical teams.

Lines 47-56: please provide an explanation related to the decision to not include in your research, condition almost overlapping to incivility, as lateral (horizontal) violence, and harassment, since lots of the cited behaviors cannot be excluded from the intent to harm, with a high level of certainty.

Response. We agree with the reviewer that this aspect needs to be clarified. Our literature review aims at capturing low intensity rude behaviors. We explained the distinction we made between low intensity rude behaviors and other terms describing more severe forms of aggression in the introduction. However, because there is obviously a lack of consistency in the literature about the definition of each term describing expression of rudeness at work, we cannot completely exclude that we did not capture some relevant work. We discuss the advantages (keeping the focus of the literature review on low intensity rude behaviors) and risks (not including a relevant publication) in the limitation section.

Methods

This section is well written and rigorously drafted. I've only some concerns about the keywords used by the authors for their literature searching. The definition and the conceptuality of "workplace incivility" in the working settings are often blurred as other "negative behaviors" that can be displayed in "abuse", "hostility", "harassment", "lateral violence", and "bullying" terms. Since many of these keywords are reported in the "concept studied" column in table 1, I think that at least these keywords should have been included in the databases searching, and later, the retrieved papers should have been assessed for the presence of inclusion/exclusion criteria.

Response: We thank the reviewer for the suggestion. We clarified in the introduction that we focused on low intensity behaviors and included this aspect as a limitation of our review, as specified in our response to the previous comment.

Moreover, the authors included the category of medical students, but not the students in other disciplines, as the student in nursing.

Response: We thank the reviewer for the comment. This decision was motivated by the fact that medical students may work with medical professionals in different disciplines while on medical internship (e.g. oncology specialists) whereas nursing students were more likely to work with nurses. We thus expected that the search term "nurses" would allow to identify the studies conducted with

nursing students (e.g. Milton, 2018). The manual search in the references was another barrier to prevent us from not identifying a relevant publication.

Lastly, the authors should clarify why they did not explore the healthcare professionals working incivilities in outpatient settings, out of hospital healthcare settings, and educational (universities) settings.

Response: We thank the reviewer for the comment. We clarified our decision to exclude classroom educational settings in the eligibility criteria section in the methods part of the manuscript.

Reviewer: 2

Broadly -in abstract focus on search strategy and findings rather than discussion, highlight key findings across all papers and state it clearly.

Response: We thank the reviewer for the comment. We modified the abstract to comply to the guidelines and added some key findings

Currently the results section is extremely long and repetitive and should be culled significantly to focus on key findings e.g. gender, age, position and specialty trends. See attachment bmjopen-2019-035471_Proof_hiAS.pdf

Response: We thank the reviewer for the comment. We added a graph to summarize the key findings regarding the characteristics of healthcare professionals most likely to be exposed to incivility. Thus, readers may get a quicker overview of key results. However, some of the dynamics suggested by the different studies – more specifically on incivility within vs between professional groups, are complex (e.g. rude behaviors expressed more towards the members of the own professional group). To avoid over-simplification about this important topic, we decided to express the full complexity of the results in a narrative way, in addition to the figure 3 and table 2.

We thank the reviewer for the additional comments included directly in the manuscript, they were very helpful.

Reviewer: 3

Abstract: The risk of bias assessment should also be included in the abstract with reference to the MERSQI tool.

Response: We added this aspect into the abstract and thank the reviewer for the comment. The use of "primary and secondary outcomes" is explicit in the abstract, but not in the methods.

Response: We clarified the study aims in method section and thank the reviewer for the comment, and adapted the abstract, so that these terms are no longer used. We agree that this prevents confusion.

Methods: It may be beneficial to specify why only English language studies were included (if for feasibility reasons for example, which is an acceptable reason). The dates of coverage for the databases should be explicitly stated. The exclusion of psychiatry patient care contexts should be in the exclusion criteria (though I wonder if it is of merit to include them, and if they are excluded - clear reasons why should be provided).

Response: We thank the reviewer for commenting on these important aspects. We clarified in the eligibility criteria section that the focus on patient incivility is predominant in studies on incivilities in psychiatric care settings, which was the reason for the exclusion. We also clarified the dates of

coverage of the literature review, and we took the opportunity to redo the search and updated the review to January 2020.

Who conducted the search of the databases? Was it a librarian? If so, this should be specified. Keywords used for the search can be included.

Response: We agree that this aspect was not clearly stated in the first version of the manuscript and thank the reviewer for the question. We now clearly mention that the first author conducted the literature search herself in the methods section. In the acknowledgement section, we clarified that the librarians gave technical support in advising the research team on the library resources available and helping the authors accessing publications that could not easily be retrieved over the usual publication repositories.

For steps 1 and 2 in the data abstraction, was a pre-made data abstraction form created? If so, was it pilot tested? It could be referenced and included as an appendix.

Response: We thank the reviewer for the question. We created a multiple-choice menu in the excel spread sheet where the publication records were managed. We did not conduct a systematic pilot testing. However, we developed the coding criteria based on the first publication records in the table, as part of an iterative process and tested this system informally to the next records.

Following the statement "Divergence in coding were resolved by discussion" - was this discussion between the two independent reviewers or the entire research team?

Response: This aspect was not described clearly in the methods section. We now specify that the authors involved in the publication selection process (SHP, VZ and SK) discussed the differences as a team. In the case these authors would not have found an agreement to resolve all the divergences on inclusion decision, we would have discussed the problematic cases with the fourth author (SY). This was, however, not necessary.

The total number of studies selected should not be reported in methods, only in results.

Response: We now report the total number of studies in the results section in the revised version of the manuscript.

Was the risk of bias assessment done independently by two investigators? If so, how were differences reconciled?

Response: We thank the reviewer for the question. The risk of bias assessment with the MERSQI scale - previously validated, was performed by one of the authors (SK), which is now stated in the manuscript. The other authors served as backup coders, this was, however, not necessary.

There is no risk of bias for the qualitative studies even though these comprised almost 20% of the included publications. You can consider using the Grades of Recommendation, Assessment, Development, and Evaluation—Confidence in the Evidence from Qualitative Reviews or another tool.

Response: We thank the reviewer for commenting on this important aspect. While working on the first version, we considered the Newcastle Ottawa scale. However, this tool did not allow us to consistently assess the different types of qualitative studies pursuing different goals. We met difficulties in finding a relevant score that could have been compared to the MERSQI score. Given the debate on assessment of the quality of qualitative studies, we decided to not formally assess the quality of the qualitative studies included. We decided not to include syntheses in the present work and thus could thus not apply the Grades of Recommendation, Assessment, Development, and

Evaluation—Confidence in the Evidence from Qualitative Reviews, although this tool would have been a possibility for another type of work. We thank the reviewer for the suggestion.

3) Results - I do not think the first paragraph in this section is required (beginning with "we will first present"), especially since the mode of synthesizing data was just outlined in the methods.
- The first section of the medical specialties paragraph was a bit hard to interpret owing to a double negative: "In the same vein, a study found that interactions with surgeons were not rated more negatively by ED physicians than were interactions with other specialists".

Response: We adapted the text accordingly to avoid the redundant information

4) Tables:

- Table 1 is very informative. I wonder if within each category of grouped studies, it is worth listing studies by MERSQI score? Is it possible to include the year of publication with the author name in column 1? I only suggest this because your discussion hypothesized that earlier studies may be of lower methodological quality, so it would be telling to see if year of publication correlated with MERSQI score.

Response: We thank the reviewer for the comment and added the date of publication in table 1. We found the idea of the reviewer to calculate a correlation between the date of publication and the MERSQI score extremely interesting. The correlation was high (above .5), supporting the suggestion that newer studies show better methodological quality. We included this result and a graph to illustrate the correlation (Figure 2) into the paper. Because the date of publication and the MERSQI score were so highly correlated, we grouped the studies by date of publication in addition to the grouping based on backgrounds of the study participants.

I see that "type of data" is a MERSQI domain and that assessment by study participant gets scored lower than objective measurement, but I wonder if this is a limitation of the MERSQI tool in its application to studies on incivility, which, as a social construct, is reliant on the perceptions of individuals? A recent scoping review of tools used to measure incivility identified the short negative acts questionnaire (S-NAQ) as the best suited tool to utilize in healthcare settings (A Scoping Review of Validated Tools to Measure Incivility in Healthcare Settings. Harris et al 2019) - but the items in this tool also seem like they are based on participant perception.

Response: We thank the reviewer for this important comment and deeper thoughts on the topic. We broadened our discussion to include related aspects and mentioned the publication by Harris et al. (2019), in the first paragraph of the limitation section of the manuscript. We agree that participants' perception is an important aspect in the research on incivility that may not be "rewarded" by the MERSQI coding scale. Interestingly, among the publications retrieved in the updated literature search presented in the revised version, we found studies relying on different methodologies than single participants' perception of incivility. For example, we discuss that the study on Heslin and colleagues (2019) who compared perspectives of different team members involved in incivility episodes and the episodes were reviewed by an independent expert panel. We may thus in the future be able to compare incivility episodes as measured based on a variety of methodologies.

2) Is there a way to correlate the MERSQI scores with high / moderate / low risk of bias? Or Good / moderate / fair quality of studies? If so, it might make it reader friendly to report how many studies fell into each category.

Response: We used the MERSQI score to assess risks of bias for each quantitative study included. We present the detail of the MERSQI scores as additional material, so that specific aspects of risks of

bias are available. We re-calculated the mean MERSQI scores and compared this value to other related medical education research.

3) There were some grammatical errors in the paper that need to be reviewed carefully.

Response: We thank the reviewer for the comment. We made our best to improve the quality of the grammar in the manuscript.

We thank the reviewer very much for the thorough and very helpful review which was inspiring and, in our views, helped us to improve the quality of the paper.

Reviewer 4:

There are a few minor typographical errors that should be corrected.

Response: We did our best to correct the grammatical errors.

Further, I would recommend including information as to the rationale for including students in your sample. Potential differences exist in the sources of incivilities experienced by student learners, as sources could be faculty versus others within the hospital.

Response: We thank the reviewer for the suggestion. We clarified that we aimed at including publications reporting data on incivility in clinical settings. We thus clarified that we included studies conducted with students only if the study explored their experiences in the clinical rather than the educational setting. We assumed that the triggers of incivility episodes may be different in potentially more stressful patient care settings vs purely educational settings (e.g. classroom). We included this aspect in the revised version.

Also, did you consult a medical librarian to develop your search criteria? If so, I would revise to include that information.

Response: We thank the reviewer for giving us the opportunity to clarify this aspect. We specified that the search was conducted by the authors (in the methods section) and also clarified the acknowledgement, stating clearly that the librarians offered support in presenting the library resources and providing access to the publications that could not be accessed otherwise by the authors.

Finally, Figure 2 on page 7 seems redundant I recommend removing from the manuscript. See attachment bmjopen-2019-035471_Proof_hi (1).pdf

Response: We thank the reviewer for the comment. We suppressed the figure mentioned by the reviewer. We added another figure showing a plot of the correlation between the date of publication and the MERSQI score. Figure 2 in the revised version provides an additional information (the regression line).

We wish to thank the reviewer for the additional and very informative comments directly in the text attached to the review.

Reviewer 5

Unfortunately, the quality of the literature included in this systematic review is poor and many unanswered questions remain.

Response: We thank the reviewer for this comment. Following the suggestion of the reviewer, we updated the literature search (including another database and extending the time period) and found

15 additional new eligible publications. As shown in Figure 2 and by the significant positive correlation between date of publication and MERSQI score, the quality of the studies in our field increased significantly in the past two years. We mention that the mean MERSQI score of the studies presented in the revised version is now extremely similar to other work in medical education.

1) The search is over 1 year old and therefore should be updated

Response: We updated the literature search up to January 2020 and could include 15 more papers.

2) The databases searched did not include Embase. To ensure key articles are not missing, Embase should be searched.

Response: We thank the reviewer for the suggestion. In the literature search conducted in January 2020, we additionally searched the full Embase data base. We thus relied on the results of five databases to revise the manuscript.

3) It may be helpful to add a table that summarizes the quantitative data, in terms of the number of positive and/or negative studies for the various predictive factors looked at.

Response: We thank the reviewer for the comment. In the revised version of the manuscript, we present the main results related to the characteristics of healthcare professionals most likely to be exposed to higher incivility level in Figure 3. Because the topic is sensitive and the results of the studies presented complex, we made the decision to also conserve a narrative presentation of the main results.

4) The presentation of the qualitative data is less clear - more discussion of these studies and their findings may strengthen the manuscript.

Response: We included the additional recently published qualitative studies in table 2. Qualitative studies are particularly informative about the situational and cultural predictors of incivility, presented in table 2.

5) When looking at gender, the authors don't appear to have separated out studies that focus on physicians from those that focus on nurses. Given that nursing is typically female dominated, the authors may want to look at the findings related to gender in only those studies that involved physicians.

Response: The studies included reported mixed findings both for physicians and nurses. We thank the reviewer for the opportunity to clarify this aspect in the results section, in the paragraph on gender differences.

6) The discussion could be strengthened by starting to explore potential strategies that could be implemented around the factors that were found to impact incivility (e.g. leadership factors).

Response: We thank the reviewer for the comment and suggestion. We included a practical implication for the organizations in the last paragraph of the discussion

VERSION 2 – REVIEW

REVIEWER	STEFANO BAMBI Careggi University Hospital, Florence - Italy
REVIEW RETURNED	23-Mar-2020

GENERAL COMMENTS	Please insert references in these sentences "Few studies focused on the impact of the countries' cultures on incivilities. Two studies, conducted with nurses, included samples from different countries. One found that the prevalence of incivilities was higher in the US compared to the Italian nurse sample. The other study compared Australian with UK nurse students and found that Australian nurse students reported more incivility.
--

REVIEWER	Amith Shetty Westmead Institute for Medical Research
REVIEW RETURNED	10-Mar-2020

GENERAL COMMENTS	Thanks for providing the opportunity to review this great piece of investigative work. My comments are provided in the file below but broadly my comments are based on rearranging the content within the structure of the manuscript. Due to the nature of the research in this field, the findings are disparate. The authors need to try and prioritise their findings into table and consider reducing the number of words in the discussion section. Nonetheless, a great piece of research and compilation of articles and insightful compendium of findings.
--

REVIEWER	Reena Pattani St Michael's Hospital, University of Toronto, Canada
REVIEW RETURNED	30-Mar-2020

GENERAL COMMENTS	The manuscript looks great and is a substantial improvement. Congratulations on this impressive work! There are a few minor typographical issues that could be addressed prior to final publication. All the best!
--